# The Associations of Erythrocyte Fatty Acids with Whole Blood Mineral Elements in Children

**DOI:** 10.3390/nu14030618

**Published:** 2022-01-30

**Authors:** Zhi Huang, Xing Liu, Ziming Li, Luwei Cui, Candi Liu, Weifeng Wang, Yuming Hu, Bo Chen

**Affiliations:** 1Key Laboratory of Phytochemical R & D of Hunan Province, School of Chemistry & Chemical Engineering, Hunan Normal University, Lu Mountain Road No. 286, Changsha 410081, China; hhsfbyzhige@hotmail.com (Z.H.); Liuxing70829@163.com (X.L.); ayanerudesu@foxmail.com (C.L.); 2Key Laboratory of Chemical Biology & Traditional Chinese Medicine Research of Ministry of Education, School of Chemistry & Chemical Engineering, Hunan Normal University, Lu Mountain Road No. 286, Changsha 410081, China; 3School of Public Health and Laboratory, Hunan University of Medicine, Jinxi Road No. 492, Huaihua 418000, China; 4The Department of Toxicology, Hunan Provincial Center for Disease Control and Prevention, Furong Road No. 450, Changsha 410005, China; hncdclzm@163.com (Z.L.); dead_blue@126.com (L.C.); a379728810@163.com (W.W.)

**Keywords:** erythrocyte, mineral elements, fatty acids, children

## Abstract

Background. Minerals play important biological roles in lipid metabolism. The primary aim of this study was to examine the relationships between erythrocyte fatty acids (FAs) levels with whole blood mineral elements concentrations among Chinese children. Methods. A cross-sectional study was conducted. A total of 435 children aged 4–7 years were recruited. Whole blood mineral elements were determined by atomic absorption spectrometry and erythrocyte FAs composition by gas chromatography-mass spectrometer. Results. There were direct correlations between Zn and C18:2n-6 (FDR corrected *p* = 0.019), total n-6 PUFAs (FDR corrected *p* = 0.034), and total PUFAs (FDR corrected *p* = 0.034). Direct correlations were found between whole blood Zn and C18:1n-9 (FDR corrected *p* = 0.035), C24:1n-9 (FDR corrected *p* = 0.023), total MUFAs (FDR corrected *p* = 0.023), and C18:2n-6 (FDR corrected *p* = 0.048) in the Cu < P50 group. In the Cu ≥ P50 group, Mg was inversely related to most FAs (All FDR corrected *p* < 0.05). In the Zn < P50 group, Cu was directly related to C24:1n-9, total MUFAs, C20:5n-3, C22:6n-3, total n-3 PUFAs, C20:4n-6, total n-6 PUFAs, total PUFAs, and total FAs (All FDR corrected *p* < 0.05). Conclusions. Whole blood Cu and Zn levels were directly linked to several FAs levels in the erythrocytes of children. The interactions of Mg, Cu, and Zn with fatty acids may affect FA metabolism, in which Mg influences FA absorption.

## 1. Introduction

Fatty acids (FAs) are major components of lipids and play an important role in energy metabolism. Polyunsaturated fatty acids (PUFAs), precursors of several intra- and extracellular messengers, mediate infection, inflammation, and hemostasis [1]. Many previous studies have confirmed that FAs are associated with metabolic and cardiovascular diseases [2,3]. The sources of human FAs depend on exogenous intake from diet and endogenous synthesis by the organism. However, essential FAs including linoleic acids (LAs, 18:2n-6) and alpha-linolenic acids (18:3n-3) cannot be endogenously synthesized by the body and must be obtained from the diet [4]. Furthermore, these FAs can further generate longer-chain omega-3 (n-3) and omega-6 (n-6) PUFAs via several enzymes, including δ-5 desaturase (∆5-D), δ-6 desaturase (∆6-D), δ-9 desaturase (∆9-D), and elongase (Elo) [4].

Minerals play important biological roles in growth, metabolism, and immunological and neurological functions [5,6]. Magnesium (Mg) acts as a necessary cofactor for several enzymes and plays a key role in glucose metabolism [7]. Iron (Fe) is associated with oxygen transport and the cytochrome enzymes reductases, catalases, and peroxidases [8]. Zinc (Zn) is required by more than 300 enzymes involved in protein synthesis [9], while copper (Cu) is important for oxidation–reduction reactions that facilitate energy production, metabolism, and detoxification [5]. Moreover, several studies have found that some mineral elements (e.g., Zn [10] and Mg [11]) are important cofactors for desaturases and elongases in endogenous FA synthesis and metabolic regulation, as well as the production of PUFAs.

Trace elements can alter FAs in human plasma, as reported in several studies. Yu et al. found that the serum levels of PUFAs are associated with the serum levels of Zn and Mg [12]. Knez et al. showed that low Zn concentrations in plasma and an altered Cu:Zn ratio are related to modified FA profiles [13]. Nevertheless, those studies did not clarify the roles of potential interactions of mineral elements in FA metabolism. Notably, the FA composition of plasma reflects recent dietary fat intake and is, thus, not a good indicator of endogenous FA production [14]. Erythrocytes better reflect long-term dietary FA intake and endogenous FA production and are widely used for this purpose [15].

The primary aim of this study was to examine the relationships between erythrocyte FA levels and whole blood calcium (Ca), Mg, Fe, Cu, and Zn concentrations, as well as the potential interactions of mineral elements in FA metabolism in Chinese children.

## 2. Materials and Methods

### 2.1. Study Population

This cross-sectional study was conducted in 2019 in a single primary school in Xiangtan, Hunan Province, located in South Central China. All children aged 4–7 years in this school were invited to participate. In total, 435 children were recruited, of whom 20 without collected blood samples were excluded.

### 2.2. Anthropometric Measurements

Anthropometric variables, including weight (kg) and height (cm), were measured using an electronic instrument. Body mass index (BMI) was calculated as weight (kg) divided by height squared (m^2^).

### 2.3. Blood Sample Collection and Analysis

Blood samples were collected by venepuncture of the cubital vein using a vacuum tube with an anticoagulant. All samples were processed within 4 h of sampling, and blood samples were divided into aliquots for FA or mineral element analysis. Moreover, the lipid profile, including triglyceride (TG), cholesterol (CHOL), high-density lipoprotein cholesterol (HDL-C), and low-density lipoprotein cholesterol (LDL-C) levels, were determined from serum samples on the same day of collection. A clinical chemistry analyzer AU680 (Beckman Coulter, Brea, CA, USA) and diagnostic kits (Fosun, Shanghai, China) were used to assess the plasma lipid parameters.

The whole blood samples for mineral element analysis were stored at −80 °C. The whole blood samples for FA analysis were centrifuged (3000 rpm, 1500× *g*, for 10 min) to separate the erythrocytes. The erythrocytes were then washed with an equal volume of saline, and the saline was removed. The remaining erythrocyte suspensions were stored in freshly butylated hydroxytoluene-treated Eppendorf vials at −80 °C.

### 2.4. Minerals Analysis

Whole blood Ca, Fe, Mg, Zn, and Cu concentrations were determined by flame atomic absorption spectrometry using the ContrAA 700 system (Analytik Jena GmbH, Jena, Germany), according to the method described by Guo et al. [16]. Recorded values were converted to mean values of triplicate samples, expressed in mg/L. Control serum samples with certified concentrations were used for quality control; the results obtained were in good agreement with the certified values.

### 2.5. Fatty Acids Determination

FA extraction and derivatization were based on a previously described procedure by Sertoglu et al. [17] with some modifications. Briefly, erythrocyte suspensions were thawed at 4 °C before the study. An internal standard working solution (IS, 100 μL), 50 μL erythrocyte suspension, and 1 mL 3 N methanolic hydrochloric acid were added to a 10 mL glass vial in the order given. The vial was closed with a screw cap to allow transmethylation to proceed at 90 °C for 1.5 h. After cooling the vials to room temperature, 2 mL hexane was added, and the vial was closed again and vortexed for 10 s. The upper (hexane) layer was transferred to a glass tube and concentrated by a drying process under a stream of nitrogen gas. Finally, the residue was dissolved in 1 mL hexane.

The samples (1 µL) were analyzed using a gas chromatography-mass spectrometer (GCMS-QP2010, Shimadzu Corp., Kyoto, Japan) and separated using an HP-88 column (dimensions, 100 m × 0.25 mm × 0.20 µm; Agilent Technologies, Santa Clara, CA, USA). The injector and detector were maintained at 230 °C. The temperature program was as follows: an initial oven temperature setting of 50 °C for 1.5 min, increase of 30 °C/min to 190 °C and holding for 5 min, followed by another increase of 8 °C/min to 230 °C and holding for 20 min. Helium was used as the carrier gas. The total analysis time was approximately 35 min. FA peaks were observed between 9.6 and 32.5 min.

Characterization and identification of FA methyl esters (FAMEs) was performed in the scan mode. Quantification was performed using the selected ion monitoring mode of the most intense fragments (saturated: *m/z* 74; monounsaturated: *m/z* 69; diunsaturated: *m/z* 81; polyunsaturated: *m/z* 79). Data acquisition and processing were performed using GC-MS Solution Software (Shimadzu). The SupelcoTM 37 Component FAME Mix (Sigma-Aldrich, St. Louis, MO, USA) was used to identify key FAMEs via chromatograms; this mixture contains the following methyl esters: butyric acid (C4:0), caproic acid (C6:0), caprylic acid (C8:0), capric acid (C10:0), undecanoic acid (C11:0), lauric acid (C12:0), tridecanoic acid (C13:0), myristic acid (C14:0), myristoleic acid (C14:1n-5), pentadecanoic acid (C15:0), pentadecenoic acid (C15:1n-5), palmitic acid (C16:0), palmitoleic acid (C16:1n-7), heptadecanoic acid (C17:0), heptadecenoic acid (C17:1n-7), stearic acid (C18:0), elaidic acid (C18:1n-9t), oleic acid (C18:1n-9), linolelaidic acid (C18:2n-6t), linoleic acid (C18:2n-6), arachidic acid (C20:0), linoleic acid (C18:3n-6), eicosenoic acid (C20:1n-9), alpha-linolenic acids (C18:3n-3), heneicosanoic acid (C21:0), eicosadienoic acid (C20:2n-6), behenic acid (C22:0), eicosatrienoic acid (C20:3 n-6), erucic acid (C22:1n-9), eicosatrienoic acid (C20:3n-3), arachidonic acid (C20:4n-6), tricosanoate acid (C23:0), docosadienoic acid (C22:2n-6), lignoceric acid (C24:0), eicosapentanoic acid (C20:5 n-3), nervonic acid (C24:1n-9), and docosahexaenoic acid (C22:6n-3). Quantification was based on calibration with methyl nonadecanoate (Aladdin, Shanghai, China) as the IS. To evaluate FA recovery, with reference to the study by Pereira et al. [18], three concentrations of tridecanoic acid (Aladdin) were added to blood samples, and the abovementioned protocol of extraction, derivatization, and quantification was followed, with a total recovery of 107.27 ± 14.15%. The results are expressed in μg/mL.

The estimated activities of desaturases and elongases are expressed as product/precursor ratios. C20:5n-3/C18:3n-3, C22:6n-3/C20:5n-3, and C20:4n-6/C18:2n-6 ratios are considered to reflect desaturase/elongase activity [18]. The C18:1n-9/C18:0 ratio represented the activity of ∆9-D. The C18:0/C16:0 and C24:1n-9/C18:1n-9 ratios were used to represent elongase activity [19].

### 2.6. Ethics Approval and Consent to Participate

All subjects provided informed consent for inclusion before participating in the study. The study was conducted in accordance with the Declaration of Helsinki, and the protocol was approved by the Ethics Committee of the Hunan Provincial Center for Disease Control and Prevention (HNCDC-BJ20190003).

### 2.7. Statistical Analysis

Continuous variables are presented as the mean (standard deviation, SD). Correlation analyses were used to explore potential relationships between different pairs of variables. For normally distributed data, Pearson’s correlation analysis was used. Spearman’s correlation analysis was used to analyze the data without a normal distribution. To explore the correlations between FA and mineral concentrations, partial correlation analyses were conducted after adjusting for age and sex. Blood Zn and Cu levels were categorized into two groups according to the 50th percentile (P50) to explore the potential interactions among the mineral elements. The associations of Cu and Zn levels with FA levels and their interaction effects on FAs in children were evaluated using a general linear model. In addition, the interactions of mineral elements with selected variables were examined with respect to the Zn and Cu status according to P50 stratification. All *p*-values of the associations or interactions with FAs or minerals were corrected for multiple testing by false discovery rate method (FDR) [20]. *p* < 0.05 was considered to indicate statistical significance. Statistical analyses were performed using SPSS software (version 13.0; SPSS, Inc., Chicago, IL, USA).

## 3. Results

### 3.1. Baseline Characteristics of the Children

Table 1 shows the baseline characteristics of the children. For children’s sex, 52.53% was boy. The mean age and BMI were 6.02 years and 15.06 kg/m^2^, respectively. The mean TG, CHOL, HDL-C, and LDL-C levels were 1.13, 4.18, 1.55, and 2.18 mmol/L, respectively. The average whole blood Ca, Mg, Fe, Cu, and Zn concentrations were 70.37, 41.89, 448.43, 0.90, and 4.64 mg/mL, respectively.

### 3.2. FAs Composition

Figure 1 shows that the main FAs in erythrocytes were saturated FAs (SFAs), ac-counting for more than 50% of the total FAs. C16:0 (191.36 μg/mL) had the highest concentration, followed by C18:0 (128.95 μg/mL). C18:1n-9 and C24:1n-9 were composed of more than 90% monounsaturated FAs (MUFAs). The mean concentrations of the essential PUFAs C18:3n-3 and C18:2n-6 were 1.64 and 78.29 μg/mL, respectively. The concentrations of C20:5n-3 and C22:6n-3 were 1.31 and 13.24 μg/mL, respectively, accounting for approximately 90% of the n-3 PUFAs. There were higher proportions (>90%) of C18:2n-6 and C20:4n-6 among n-6 PUFAs. Therefore, the associations of these main FAs with minerals were the focus of our subsequent investigation.

### 3.3. Correlations of Mineral Element and FA Concentrations with Age, BMI, and Plasma Lipid Parameters

Correlation analyses revealed inverse relationships of the Ca level with age (FDR corrected *p* < 0.001) and of the Cu level with the HDL-C level (FDR corrected *p* = 0.011). There were direct correlations between CHOL and Zn levels (FDR corrected *p* = 0.044). Most of the FAs had negative associations with age (all FDR corrected *p* < 0.05). There were direct correlations of CHOL with C22:6n-3 levels (FDR corrected *p* = 0.016) and total n-3PUFAs (FDR corrected *p* = 0.032). No statistically relevant associations were found between BMI, TG, or LDL-C and mineral element or FA levels (Table 2).

### 3.4. Correlation of Mineral Element with FA

Figure 2 shows significant positive correlations of blood Mg with Fe (FDR corrected *p* = 0.017), Cu (FDR corrected *p* < 0.001), and Zn (FDR corrected *p* = 0.006), and a significant inverse relationship between Cu and Fe (FDR corrected *p* = 0.032). Table 3 lists the partial correlation coefficients adjusted for age and sex of whole blood mineral elements with erythrocyte FAs in children. There were direct correlations between Zn and C18:2n-6 (FDR corrected *p* = 0.019), total n-6 PUFAs (FDR corrected *p* = 0.034), and total PUFAs (FDR corrected *p* = 0.034). No statistically significant relationships were observed among the whole blood status of Ca, Mg, Fe, and Cu with any of the FAs (Table 3).

### 3.5. The Role of the Potential Interactions of Mineral Elements in FA Metabolism

As shown in Table 4, there were weak correlations of Zn with C20:5n-3, C18:2n-6, total n-6PUFAs, and total PUFAs levels (all FDR corrected *p* = 0.053), as well as between Cu and C22:6n-3, total n-3PUFAs, and C20:4n-6 levels (all FDR corrected *p* = 0.052), but there were no interactions of FAs with Zn and Cu using our linear model.

However, according to P50 stratification of Zn and Cu status, direct correlations were found between whole blood Zn and C18:1n-9 (FDR corrected *p* = 0.035), C24:1n-9 (FDR corrected *p* = 0.023), total MUFAs (FDR corrected *p* = 0.023), and C18:2n-6 (FDR corrected *p* = 0.048) in the Cu < P50 group. Moreover, there were weak relationships between Zn and total n-6 PUFAs, total PUFAs, and total FAs (All FDR corrected *p* = 0.052). No statistically relevant association was found between Ca, Mg, Fe, or Cu status and FAs. In the Cu ≥ P50 group, Mg was inversely related to C16:0, C18:0, total SFAs, C18:1n-9, C24:1n-9, total MUFAs, C22:6n-3, total n-3 PUFAs, C18:2n-6, C20:4n-6, total n-6 PUFAs, total PUFAs, and total FAs (all FDR corrected *p* < 0.05). No statistically significant relationships were found between Ca, Fe, Cu, or Zn status and any of the FAs (Figure 3a and Appendix A)

In the Zn < P50 group, Ca, Mg, Fe, and Zn did not show any association with any of the FAs. Cu was directly related to C24:1n-9, total MUFAs, C20:5n-3, C22:6n-3, total n-3 PUFAs, C20:4n-6, total n-6 PUFAs, total PUFAs, and total FAs (all FDR corrected *p* < 0.05). In the Zn ≥ P50 group, Ca, Mg, Fe, Cu, and Zn had no correlations with FAs (Figure 3b and Appendix A).

### 3.6. Potential Mechanisms of the Mineral Elements on FA Metabolism

Figure 4 shows the metabolic pathways associated with the main FAs [19]. A series of enzymes are needed for FA biosynthesis. These enzymes are involved in the desaturation/elongation process of C18:3n-3 and C18:2n-6 to C22:6n-3 and C20:4n-6. The C20:5n-3/C18:3n-3, C22:6n-3/C20:5n-3, and C20:4n-6/C18:2n-6 ratios are considered to reflect desaturase/elongase activity. In this study, a significant positive correlation was found between the C20:4n-6/C18:2n-6 ratio and Cu (FDR corrected *p* = 0.002). On the other hand, ∆9-D is responsible for the production of C18:1n-9 from C18:0. Cu (FDR corrected *p* = 0.038) and Zn (FDR corrected *p* = 0.006) showed significant positive correlations with the C18:1n-9/C18:0 ratio. In addition, elongase was involved in the production of C18:0 from C16:0. Thus, the C18:0/C16:0 ratio can be used as an indicator of elongase activity that is inversely correlated with Cu (FDR corrected *p* = 0.049) (Table 5).

## 4. Discussion

This cross-sectional study evaluated the associations of erythrocyte FA composition with whole blood mineral element concentrations among Chinese children. In this study, the composition of FAs in erythrocytes depended mainly on the SFAs, which accounted for more than 50% of FAs. Additionally, high proportions of C18:1n-9 and C24:1n-9 in MUFAs, C22:6n-3 and C20:4n-6 in n-3 PUFAs, and C18:2n-6 and C20:4n-6 in n-6 PUFAs were found in the children, similar to those found in other populations (e.g., pregnant women [21], adults [2], and the elderly [22]). The mean concentrations of the essential PUFAs C18:3n-3 and C18:2n-6 were 1.64 and 78.29 μg/mL, respectively, which was lower than the values reported by Araujo et al. (9.49 µg/g for C18:3n-6 and 225.68 µg/g for C18:2n-6) [21], but higher than that reported by Gunes et al. (43.15 μg/mL for C18:2n-6) [14].

Furthermore, the current study showed that alterations in the erythrocyte FA composition were related to whole blood mineral element status, particularly those of Zn, which were positively associated with C18:2n-6, total n-6 PUFAs, and total PUFAs by correlation analysis. Moreover, there were weak correlations of Zn with several PUFAs, as well as between Cu and several PUFAs by general linear model after FDR correction. Several previous studies have confirmed that Zn is involved in PUFA metabolism, especially for LA:DGLA (C18:2n-6:C20:3n-6) [23,24,25]. Knez et al. reported that the plasma Cu level is directly related to oleic acid (C18:1 n-9) and alpha-linolenic acid (C18:3 n-3) [13]. Mahfouz et al. observed that Mg deficiency reduces the rate of conversion of linoleic acid (C18:2n-6) to arachidonic acid (C20:4n-6) [11]. In addition, our study found that there was a direct correlation between CHOL and Zn. Knez et al. [13] reported plasma Zn was directly and Cu:Zn ratio inversely associated with LDL-C/HDL-C and CHOL/HDL-C. On the other hand, C22:6n-3 and total n-3PUFAs had association with CHOL in this study. These results hinted that Zn may involve in lipid metabolism by altering FAs composition.

There were no interaction effects of Zn and Cu on FAs according to our general linear model. Nevertheless, our results showed the Zn level was directly related to C18:1n-9, C24:1n-9, total MUFAs, and C18:2n-6 in the Cu < P50 group, and there were weak relationships between Zn and total n-6 PUFAs, total PUFAs, and total FAs. However, no statistically significant relationships were observed between Zn and FA levels in the Cu ≥ P50 group. Additionally, the direct correlations between Cu and total MUFA, total n-3 PUFA, total n-6 PUFA, total PUFA, and total FA levels in the Zn < P50 group, whereas no statistically significant relationships were observed between Cu status and any of the FAs in the Zn ≥ P50 group. Mg was inversely related to most of the FAs in the Cu ≥ P50 groups. These results indicate that Cu may modify FA metabolism if there is a Zn deficiency, and that Zn may modify FA metabolism if there is a Cu deficiency, whereas Mg may alter FA metabolism in the presence of higher levels of Cu. Thus, there may be interactions among Cu, Zn, and Mg with respect to FA metabolism. An antagonistic relationship between Zn and Cu has been reported in some studies [26,27]. In the current study, no statistically significant relationships were observed between Cu and Zn levels in whole blood; however, positive correlations of the blood Mg level with Cu and Zn levels were found.

Our results showed that minerals are correlated with several long-chain FAs (e.g., C20:5n-3, C22:6n-3, and C20:4n-6). These FAs can be consumed from food in minor amounts; however, their status in the body mainly reflects endogenous synthesis [28]. The endogenous synthesis of these FAs depends primarily on a series of enzymes, including desaturase and elongase. Zn is a cofactor for desaturase (e.g., ∆6D and ∆5D enzymes) [4]. Cu plays an important role in the metabolic pathways of all FA desaturases, as a component of cupoprotein enzymes, resulting in increased desaturase activity [29]. In this study, Cu and Zn showed a significant positive correlation with the C18:1n-9/C18:0 ratio, representing by ∆9-D activity. Moreover, a significant positive correlation was found between the Cu level and C20:4n-6/C18:2n-6 ratio, representing desaturase/elongase activity. However, the Cu level was inversely related to the C18:0/C16:0 ratio, representing elongase activity, which may be attributed to the Fe status. Fe is also a component of a rate-limiting enzyme involved in FA synthesis [30]. A significant inverse relationship between the blood Cu and Fe level was found in this study. Therefore, the associations of FAs with Cu and Zn may involve the endogenous synthesis of FAs.

In addition, our results showed a direct correlation between C18:2n-6 and Zn levels. C18:2n-6, an essential FA for humans, must be obtained from the diet. Previous studies have reported a role of Zn in the absorption of linoleic acid, in which lower levels of Zn produce lower levels of linoleic acid [31]. Our study showed Mg was inversely related to most of the FAs (including C18:2n-6) in the Cu ≥ P50 group; however, no correlations were found between Mg and the desaturation/elongation process. These results suggest that Mg may alter the process of FA absorption.

The strengths of our study include the use of erythrocytes to reflect long-term dietary FA intake and of FA endogenous production to assess the status of individual FAs. Moreover, the potential interactions and mechanisms of mineral elements in FA metabolism was a key focus of this study. A potential limitation of the study is the cross-sectional design, which prevents clarification of the causal correlations between mineral elements and FAs. Furthermore, several factors (e.g., dietary, lifestyle, and genetic factors) [32] involved in lipid metabolism according to previous studies as potential confounding factors were not focused, which may disturb the correlation of FAs with minerals. In addition, this study included only children; thus, our findings may not be representative of the general population.

## 5. Conclusions

In this cross-sectional study, whole blood Cu and Zn levels were directly linked to several FAs levels in the erythrocytes of children. The interactions of Mg, Cu, and Zn with fatty acids may affect FA metabolism, in which Mg influences FA absorption. These results provide evidence for precision nutrition intervention in children.

## Figures and Tables

**Figure 1 nutrients-14-00618-f001:**
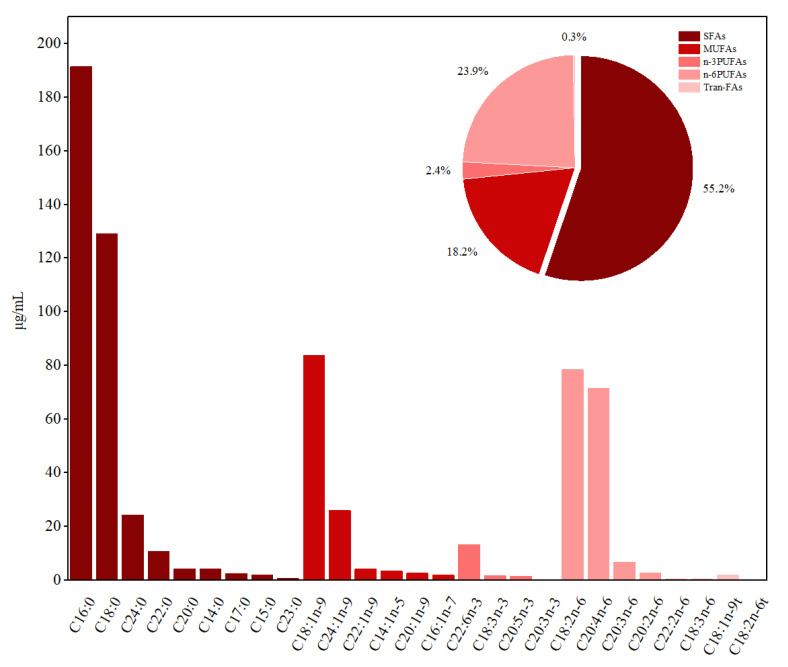
The concentration and constituent ratio of FAs in erythrocyte among children. Data for the concentration of FAs are presented as mean. Abbreviations: SFAs—saturated fatty acids; MUFAs— monounsaturated fatty acids; PUFAs—polyunsaturated fatty acids; Tran-FAs—tran-fatty acids.

**Figure 2 nutrients-14-00618-f002:**
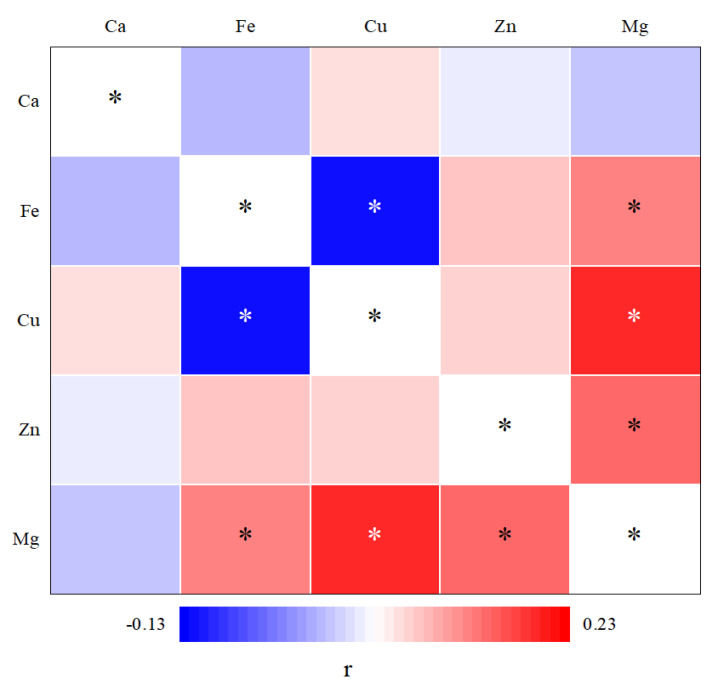
The correlation of different whole blood mineral elements. r: correlation coefficients. *: *p* < 0.05, the *p*-value was adjusted using false discovery rate method for multiple testing.

**Figure 3 nutrients-14-00618-f003:**
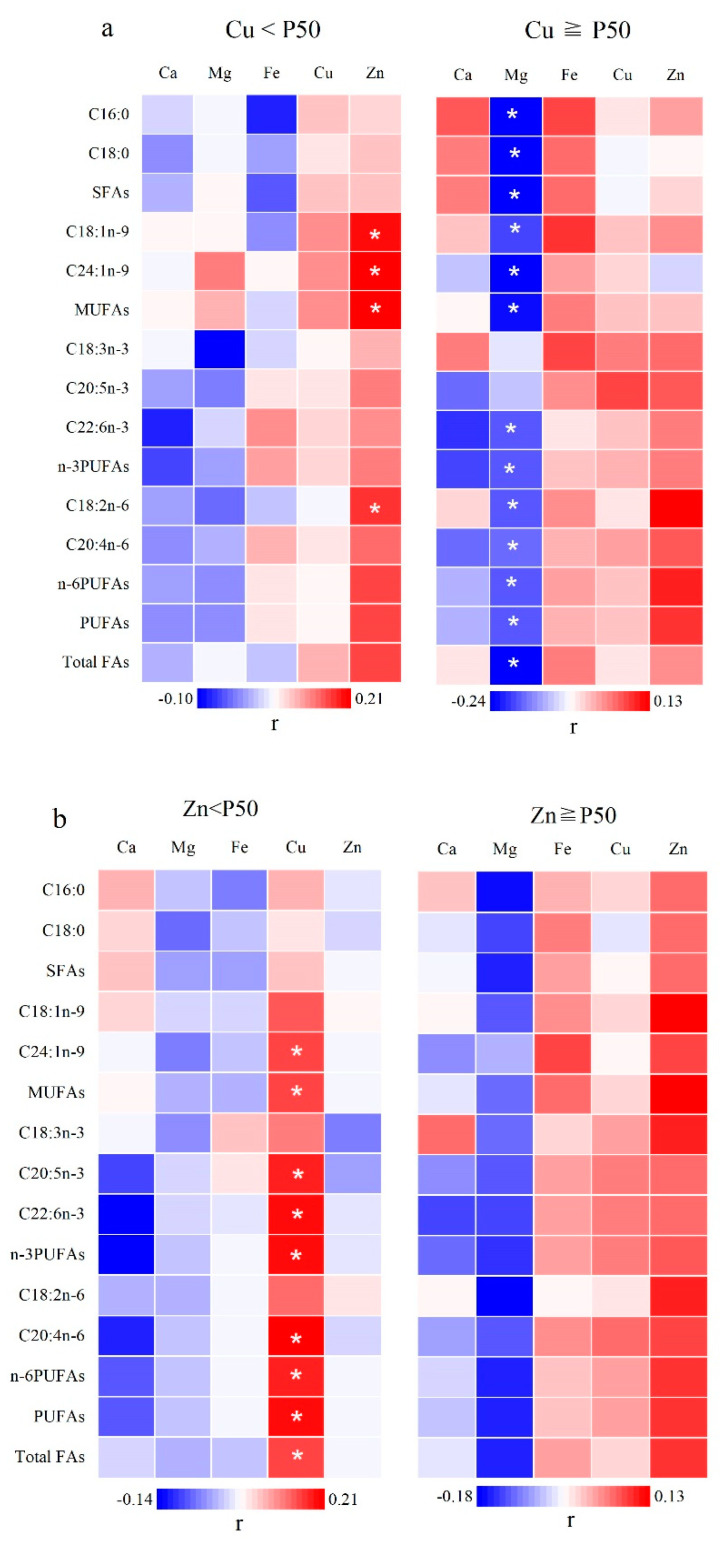
Partial correlation analysis of mineral elements with erythrocyte FAs in children. (**a**) According to P50 stratification of whole blood Cu status. (**b**) According to P50 stratification of whole blood Zn status. r: partial correlation coefficients adjusting for age and sex. * *p* < 0.05, the *p*-value was adjusted using false discovery rate method for multiple testing. Abbreviations: SFAs—saturated fatty acids; MUFAs—monounsaturated fatty acids; PUFAs—polyunsaturated fatty acids; P50—the 50th percentile.

**Figure 4 nutrients-14-00618-f004:**
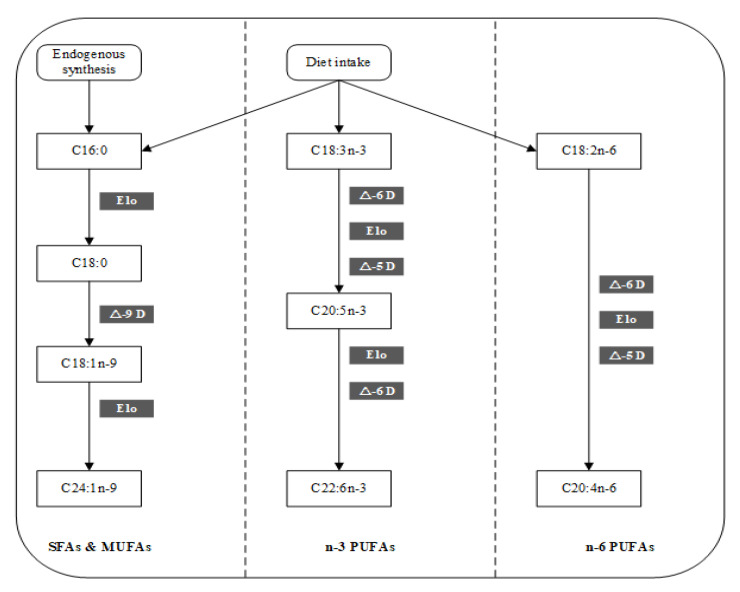
Pathways of main fatty acid metabolism. Abbreviations: SFAs—saturated fatty acids; MUFAs—monounsaturated fatty acids; PUFAs—polyunsaturated fatty acids; Elo—elongase; D—desaturase.

**Table 1 nutrients-14-00618-t001:** Baseline characteristics of children (*n* = 415).

Characteristics	Mean (SD) or N (%)
Sex, N (%)	
Boy	218 (52.53)
Girl	197 (47.47)
Age, years	6.02 (0.64)
BMI	15.06 (1.78)
Plasma lipid parameters, mmol/L	
TG	1.13 (0.73)
CHOL	4.18 (0.73)
HDL-C	1.55 (0.28)
LDL-C	2.18 (0.57)
Mineral elements, mg/L	
Ca	70.37 (8.44)
Mg	41.89 (5.50)
Fe	448.43 (85.29)
Cu	0.90 (0.13)
Zn	4.64 (0.85)

Abbreviations: SD—standard deviation; BMI—body mass index; TG—triglycerides; CHOL—cholesterol; HDL-C—high-density lipoprotein cholesterol; LDL-C—low-density lipoprotein cholesterol.

**Table 2 nutrients-14-00618-t002:** Correlation of mineral element and erythrocyte FAs with age, BMI, and plasma lipid parameters in children.

	Age			BMI			TG			CHOL			HDL-C			LDL-C		
	r	*p*	*p* _FDR_ ^a^	r	*p*	*p* _FDR_ ^a^	r	*p*	*p* _FDR_ ^a^	r	*p*	*p* _FDR_ ^a^	r	*p*	*p* _FDR_ ^a^	r	*p*	*p* _FDR_ ^a^
Mineral elements																		
Ca	−0.270	<0.001	<0.001	0.059	0.233	0.388	0.016	0.742	0.927	−0.034	0.484	0.547	−0.036	0.470	0.470	−0.018	0.709	0.709
Mg	−0.049	0.317	0.573	0.074	0.133	0.333	−0.054	0.277	0.691	0.073	0.140	0.350	−0.091	0.065	0.163	0.096	0.050	0.125
Fe	0.047	0.344	0.573	0.080	0.105	0.333	0.092	0.061	0.305	−0.030	0.547	0.547	−0.055	0.261	0.327	0.055	0.266	0.333
Cu	−0.004	0.933	0.933	0.020	0.686	0.686	−0.034	0.484	0.807	0.056	0.258	0.430	−0.149	0.002	0.011	0.069	0.163	0.272
Zn	−0.013	0.790	0.933	0.026	0.600	0.686	−0.003	0.955	0.955	0.129	0.009	0.044	0.070	0.154	0.257	0.096	0.050	0.125
FAs																		
C16:0	−0.159	0.001	0.017	−0.044	0.376	0.497	−0.002	0.968	0.968	0.049	0.320	0.534	0.011	0.826	0.826	0.051	0.304	0.627
C18:0	−0.101	0.040	0.054	−0.022	0.652	0.652	−0.014	0.775	0.830	0.028	0.565	0.771	0.045	0.361	0.493	0.025	0.618	0.772
SFAs	−0.132	0.007	0.030	−0.039	0.431	0.497	−0.020	0.690	0.796	0.044	0.376	0.564	0.032	0.516	0.608	0.044	0.374	0.627
C18:1n-9	−0.131	0.007	0.030	−0.041	0.402	0.497	0.101	0.039	0.195	0.015	0.764	0.854	−0.016	0.746	0.799	−0.002	0.965	0.965
C24:1n-9	−0.007	0.886	0.886	−0.042	0.391	0.497	−0.053	0.282	0.490	−0.011	0.819	0.854	0.100	0.043	0.232	−0.040	0.418	0.627
MUFAs	−0.084	0.088	0.102	−0.035	0.483	0.517	0.061	0.215	0.490	0.009	0.854	0.854	0.031	0.527	0.608	−0.018	0.707	0.816
C18:3n-3	−0.097	0.049	0.061	0.061	0.217	0.443	0.236	0.000	0.195	−0.009	0.852	0.854	−0.059	0.231	0.366	−0.054	0.276	0.627
C20:5n-3	−0.079	0.106	0.114	0.068	0.167	0.443	0.106	0.031	0.195	0.087	0.075	0.188	0.103	0.036	0.232	0.013	0.799	0.856
C22:6n-3	−0.105	0.032	0.048	−0.083	0.091	0.443	0.032	0.513	0.700	0.160	0.001	0.016	0.091	0.064	0.232	0.113	0.021	0.316
n-3PUFAs	−0.115	0.019	0.042	−0.057	0.243	0.443	0.062	0.207	0.490	0.140	0.004	0.032	0.082	0.097	0.232	0.087	0.076	0.570
C18:2n-6	−0.107	0.030	0.048	−0.062	0.210	0.443	0.066	0.181	0.490	0.078	0.114	0.244	0.070	0.156	0.292	0.035	0.471	0.643
C20:4n-6	−0.112	0.022	0.042	−0.070	0.152	0.443	0.034	0.486	0.700	0.111	0.024	0.120	0.085	0.082	0.232	0.063	0.201	0.627
n-6PUFAs	−0.112	0.022	0.042	−0.065	0.184	0.443	0.052	0.294	0.490	0.096	0.052	0.155	0.079	0.108	0.232	0.049	0.323	0.627
PUFAs	−0.114	0.021	0.042	−0.065	0.186	0.443	0.053	0.278	0.490	0.102	0.038	0.144	0.080	0.104	0.232	0.054	0.275	0.627
Total FAs	−0.130	0.008	0.030	−0.055	0.266	0.443	0.024	0.619	0.774	0.067	0.176	0.330	0.057	0.244	0.366	0.040	0.415	0.627

^a^ The *p*-value was adjusted using false discovery rate method for multiple testing. Abbreviations: BMI—body mass index; TG—triglycerides; CHOL—cholesterol; HDL-C—high-density lipoprotein cholesterol; LDL-C—low-density lipoprotein cholesterol; SFAs—saturated fatty acids; MUFAs—monounsaturated fatty acids; PUFAs—polyunsaturated fatty acids.

**Table 3 nutrients-14-00618-t003:** Partial correlation analysis of mineral elements with erythrocyte FAs in children.

FAs	Ca			Mg			Fe			Cu			Zn		
	r	*p* ^a^	*p* _FDR_ ^b^	r	*p* ^a^	*p* _FDR_ ^b^	r	*p* ^a^	*p* _FDR_ ^b^	r	*p* ^a^	*p* _FDR_ ^b^	r	*p* ^a^	*p* _FDR_ ^b^
C16:0	0.057	0.252	0.596	−0.094	0.057	0.171	−0.008	0.879	0.896	0.042	0.393	0.454	0.054	0.276	0.318
C18:0	0.020	0.679	0.858	−0.105	0.032	0.171	0.025	0.607	0.771	0.001	0.981	0.981	0.017	0.732	0.732
SFAs	0.030	0.542	0.825	−0.099	0.044	0.171	0.006	0.896	0.896	0.025	0.614	0.658	0.037	0.448	0.480
C18:1n-9	0.029	0.550	0.825	−0.066	0.184	0.246	0.033	0.509	0.763	0.067	0.172	0.258	0.110	0.026	0.077
C24:1n-9	−0.020	0.686	0.858	−0.059	0.229	0.246	0.038	0.447	0.763	0.062	0.209	0.285	0.060	0.224	0.280
MUFAs	0.011	0.819	0.869	−0.068	0.169	0.246	0.033	0.506	0.763	0.068	0.170	0.258	0.096	0.050	0.091
C18:3n-3	0.046	0.352	0.661	−0.068	0.169	0.246	0.044	0.369	0.763	0.073	0.137	0.258	0.081	0.101	0.138
C20:5n-3	−0.083	0.093	0.348	−0.049	0.324	0.324	0.045	0.363	0.763	0.108	0.028	0.103	0.103	0.036	0.091
C22:6n-3	−0.128	0.009	0.137	−0.062	0.211	0.246	0.036	0.466	0.763	0.118	0.017	0.083	0.089	0.072	0.108
n-3PUFAs	−0.115	0.019	0.144	−0.071	0.152	0.246	0.039	0.428	0.763	0.120	0.015	0.083	0.095	0.054	0.091
C18:2n-6	−0.008	0.869	0.869	−0.097	0.048	0.171	0.020	0.692	0.799	0.051	0.301	0.376	0.158	0.001	0.019
C20:4n-6	−0.085	0.085	0.348	−0.060	0.224	0.246	0.044	0.370	0.763	0.128	0.009	0.083	0.114	0.020	0.075
n-6PUFAs	−0.054	0.278	0.596	−0.078	0.111	0.241	0.034	0.494	0.763	0.097	0.048	0.120	0.137	0.005	0.034
PUFAs	−0.061	0.213	0.596	−0.078	0.112	0.241	0.035	0.482	0.763	0.101	0.040	0.120	0.133	0.007	0.034
Total FAs	−0.010	0.834	0.869	−0.095	0.053	0.171	0.025	0.617	0.771	0.070	0.156	0.258	0.096	0.052	0.091

^a^ Adjustment: age and sex. ^b^ The *p*-value was adjusted using false discovery rate method for multiple testing. Abbreviations: SFAs—saturated fatty acids; MUFAs—monounsaturated fatty acids; PUFAs—polyunsaturated fatty acids.

**Table 4 nutrients-14-00618-t004:** The interactions of Zn and Cu in FA metabolism among children by general linear model.

FAs	Zn ^a^		Cu ^b^		Interaction ^c^	
	*p*	*p* _FDR_ ^d^	*p*	*p* _FDR_ ^d^	*p*	*p* _FDR_ ^d^
C16:0	0.417	0.521	0.473	0.546	0.634	0.634
C18:0	0.888	0.888	0.863	0.863	0.349	0.476
SFAs	0.728	0.780	0.575	0.616	0.493	0.569
C18:1n-9	0.148	0.222	0.313	0.391	0.428	0.535
C24:1n-9	0.584	0.674	0.249	0.373	0.047	0.454
MUFAs	0.223	0.304	0.276	0.377	0.192	0.454
C18:3n-3	0.127	0.219	0.230	0.373	0.294	0.454
C20:5n-3	0.014	0.053	0.078	0.195	0.627	0.634
C22:6n-3	0.088	0.189	0.010	0.052	0.262	0.454
n-3PUFAs	0.060	0.150	0.010	0.052	0.264	0.454
C18:2n-6	0.004	0.053	0.187	0.351	0.267	0.454
C20:4n-6	0.023	0.070	0.006	0.052	0.303	0.454
n-6PUFAs	0.009	0.053	0.028	0.084	0.275	0.454
PUFAs	0.011	0.053	0.024	0.089	0.269	0.454
Total FAs	0.131	0.219	0.143	0.306	0.274	0.454

Blood Zn and Cu were categorized into two groups according the 50th percentile (P50) in general linear model. ^a^ Main effect of Zn with FAs; ^b^ main effects of Cu with FAs; ^c^ interaction effects of Cu and Zn with FAs; ^d^ the *p*-value was adjusted using false discovery rate method for multiple testing. Abbreviations: SFAs—saturated fatty acids; MUFAs—monounsaturated fatty acids; PUFAs—polyunsaturated fatty acids.

**Table 5 nutrients-14-00618-t005:** Partial correlation coefficients of mineral elements with several indices of desaturases and elongases in erythrocyte among children.

Indices	Ca			Mg			Fe			Cu			Zn		
	r	*p* ^a^	*p* _FDR_ ^b^	r	*p* ^a^	*p* _FDR_ ^b^	r	*p* ^a^	*p* _FDR_ ^b^	r	*p* ^a^	*p* _FDR_ ^b^	r	*p* ^a^	*p* _FDR_ ^b^
C18:0/C16:0	−0.088	0.075	0.120	−0.064	0.192	0.684	0.104	0.035	0.207	−0.111	0.024	0.049	−0.072	0.143	0.429
C24:1n-9/C18:1n-9	−0.086	0.080	0.120	−0.030	0.545	0.845	0.041	0.405	0.810	−0.014	0.782	0.782	−0.004	0.931	0.931
C18:1n-9/C18:0	0.026	0.599	0.599	0.059	0.228	0.684	−0.007	0.884	0.884	0.122	0.013	0.038	0.162	0.001	0.006
C20:5n-3/18:3n-3	−0.115	0.031	0.120	−0.011	0.845	0.845	0.016	0.768	0.884	0.069	0.200	0.300	0.057	0.284	0.569
C22:6n-3/C20:5n-3	−0.114	0.059	0.120	−0.018	0.765	0.845	−0.036	0.550	0.826	−0.021	0.729	0.782	0.005	0.929	0.931
C20:4n-6/C18:2n-6	−0.071	0.150	0.180	0.021	0.665	0.845	0.079	0.106	0.318	0.177	<0.001	0.002	0.031	0.535	0.802

^a^ Adjustment: age and sex. ^b^ The *p*-value was adjusted using false discovery rate method for multiple testing.

## Data Availability

The data presented in this study are available on request from the corresponding author. The data are not publicly available due to privacy.

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
