# Peer review of "The Associations of Erythrocyte Fatty Acids with Whole Blood Mineral Elements in Children"

_nutrients, 2022, doi:10.3390/nu14030618_

Round 1

Reviewer 1 Report

The authors should be tempted to determine not only fatty acids but also the lipids which these acids are composed of. Is only these 5 metal ions playing an important role in lipid metabolism, or is this choice caused by other factors?
The authors do not talk about the other factors involved in lipid metabolism only about these 5 metal ions.
Tables should be preformatted, mean and SD should be clearly described.

Statistically significant differences should be clearly marked

Author Response

Dear Editors and Reviewers

We are very grateful for your advices and comments about our manuscript. Those were helpful to improve our paper. We had made a point-by-point revision and hope to meet with approval.  Revised portion are marked in red in the paper.The responds to the reviewer’s comments are as following:

1.The authors should be tempted to determine not only fatty acids but also the lipids which these acids are composed of.

Respond: We are grateful for this advice. In our section of results, the correlation of minerals with plasma lipid had been explored and the result was showed in table 2. There were inverse relationships of Cu with HDL-C. There were direct correlations between CHOL and Zn. However, there were no significant correaltion of Ca, Mg and Fe with plasma lipid parameters. Moreover, we added interpretation for the associations of minerals with plasma lipid parameters in the second paragraph of discussion.

2.Is only these 5 metal ions playing an important role in lipid metabolism, or is this choice caused by other factors? The authors do not talk about the other factors involved in lipid metabolism only about these 5 metal ions.

Respond: We are very thankful for reviewer’s this suggestion. The lipid status may had associaitions with the changes in diet, lifestyle habits and genetics according to the reports of several previous studies. However, information for dietary, lifestyle and genetic factors of children in our study were not collected. It was very regretful to not explore the effects of these factors on the association of minerals with FAs. This advice also give us some good ideals in our further study. Therefore, we added relative descriptions in section of limitation. Nevertheless, to explore potential confounding effects for the correlation of FAs with minerals, partial correlation tests were used, adjusted for some demographic factors (including age and sex).

3.Tables should be preformatted, mean and SD should be clearly described. Statistically significant differences should be clearly marked

Respond: We are very sorry for our improper expression. We revized the tables in our study. Mean and SD had been clearly described in table 1. Statistically significant differences had been clearly marked in all Tables and Figures as notes.

Reviewer 2 Report

The authors examined the association between erythrocyte FA composition and whole blood mineral elements in children. The study is well designed and conducted however, data presentation, language, rational and conclusions are not good.

English should be massively improved. As it stand the text precludes proper understanding of the presented content.

page 4 lines 161 - 180: description of the presented data is poor. Significant differences should be described in the text as : significantly higher or lower nit only higher  or lower.

page 4 lines 169 - 171: this sentence shoold be incorporated into the section 3.3 describing correlations; this is not a part of baseline characteristics.

Why in table 1 the authors present boys and girls separately ? The residual analyses were done only in the whole sample. Despite this the authors call the study cross-sectional study.

In Table 2 and in corresponding text I would suggest to replace ''biochemical'' with ''plasma lipid'' parameters

Rational for data presented in Table 4 and description of the abbreviations in Table 4 are missing.

P-Values should be corrected for multiple comparison and accordingly it needs to be indicated below which P value the data are significant.

Discussion should be shorter with less repetition  and more interpretation of the results.

Page 11: lane 281 arachidonic acid (C18:3n-6)- please correct.

Page 12, lane 342-343: this final sentence is not clear

The discussion and conclusion contain a lot of overinterpretation of the data. The design of the study precludes several conclusions presented by the authors.

Author Response

Dear Editors and Reviewers

We are very grateful for your advices and comments about our manuscript. Those were helpful to improve our paper. We had made a point-by-point revision and hope to meet with approval. According to your comments, we revized the section of “Discussion and conclusion”. Several overinterpretation of the data were deleted. Furtheremore, the data presentation in section of results was improved. The table 1 was divided into two sections. A figure presented the composition of FAs was added (Figure 1). Revised portion are marked in red in the paper.The responds to the reviewer’s comments are as following:

The authors examined the association between erythrocyte FA composition and whole blood mineral elements in children. The study is well designed and conducted however, data presentation, language, rational and conclusions are not good.

1.English should be massively improved. As it stand the text precludes proper understanding of the presented content.

Respond: We are very sorry for our poor English level. To improve questions of language in our manuscript, we had invited two professional editors both native speakers of English to check our manuscripts again and hope that the correction will meet with approval.

2.page 4 lines 161 - 180: description of the presented data is poor. Significant differences should be described in the text as : significantly higher or lower nit only higher or lower.

Respond: We are very sorry for our poor expression. We had rewritten this section. The section 3.1 was divided into two section. The section 3.1 showed the baseline characteristics of children. The concentration and constituent ratio of fatty acids in erythrocyte among children was showed in figure (Section 3.2 and Figure 1). Some poor description had been rewrittern or deleted.

  1. page 4 lines 169 - 171: this sentence shoold be incorporated into the section 3.3 describing correlations; this is not a part of baseline characteristics.

Respond: According to reviewer’s proposal, the description for the correlation of mineral elements with FAs had been incorporated into the section 3.3

  1. Why in table 1 the authors present boys and girls separately ? The residual analyses were done only in the whole sample. Despite this the authors call the study cross-sectional study.

Respond: According to reviewer’s suggestion, we revised the presentation. The baseline characteristics were done only in the whole sample and not described according to sex. The revised reslut was showed in table 1.

  1. In Table 2 and in corresponding text I would suggest to replace ''biochemical'' with ''plasma lipid'' parameters

Respond: According to reviewer’s suggestion, ''biochemical'' parameters was replaced  with ''plasma lipid''.

  1. Rational for data presented in Table 4 and description of the abbreviations in Table 4 are missing.P-Values should be corrected for multiple comparison and accordingly it needs to be indicated below which P value the data are significant.

Respond: We are very sorry for our negligence. We revised the Table 4.The abbreviations in table 4 were noted. PZn: P-value for main effects of Zn with FAs; PCu: P-value for main effects of Cu with FAs; PZn×Cu: P-value for interaction effects of Cu and Zn with FAs. P-Values were corrected and P < 0.05 was considered to indicate statistical significance.

  1. Discussion should be shorter with less repetition and more interpretation of the results.

Respond: According to reviewer’s suggestion, we further revised section of discussion. Several descriptions of repetition were removed.

  1. Page 11: lane 281 arachidonic acid (C18:3n-6)- please correct.

Respond: We are very sorry for our wrong expression. The arachidonic acid (C18:3n-6) was revised into arachidonic acid (C20:4n-6).

  1. Page 12, lane 342-343: this final sentence is not clear

Respond: We rewrote this sentence. In addition, elongase was involved in the production of C18:0 from C16:0. Thus, the C18:0/C16:0 ratio can be used as an indicator of elongase activity that is inversely cor-related with Cu and directly correlated with Fe

  1. The discussion and conclusion contain a lot of overinterpretation of the data. The design of the study precludes several conclusions presented by the authors.

Respond: We are very sorry for our improper expression. The conclusion had been rewritten. Some content of repetition had been removed.

Round 2

Reviewer 2 Report

The authors improved the manuscript, but  one of my previous important comments was not considered.

I asked that p-values in the Tables 2,3 and 4 and in the figs 2 and 3 with a lot of variables tested for their correlattions are corrected for multiple comparisons. This can be done in a statistic program using option correction for multiple comparison or simply by deviding (Bonferroni) 0.05 with number of comparisons per column; in this case in table 2 results for minerals are significant when P value is lower than 0.05/5 = 0.01 and for FAs 0.05/15 = 0.003. Similarly should be done in other Tables and Figs. I would suggest presenting exact  P-values not * and indicate in the table /Fig legend the P-value under which results are considered significant.

It is also importnat to stress that all correlation coefficients are low or very low . This in combination with corrections for multiple comparisons which may indicate that some correlations are not significant can markedly change the conclusion of the paper.

The authors concluded that their results provide new biomarkers. This is not possible to conclude based on the very week correlations and without corrections for multiple comparisons.

Why the study is called cross-sectional?

Lane 261: Figure 3 should be Figure 4. Please correct.

Author Response

Dear Editors and Reviewers

We are very grateful for your advices and comments about our manuscript. Those were helpful to improve our paper. We had made a point-by-point revision and hope to meet with approval. Revised portion are marked in red in the paper. The responds to the reviewer’s comments are as following:

The authors improved the manuscript, but one of my previous important comments was not considered.

I asked that p-values in the Tables 2,3 and 4 and in the figs 2 and 3 with a lot of variables tested for their correlattions are corrected for multiple comparisons. This can be done in a statistic program using option correction for multiple comparison or simply by deviding (Bonferroni) 0.05 with number of comparisons per column; in this case in table 2 results for minerals are significant when P value is lower than 0.05/5 = 0.01 and for FAs 0.05/15 = 0.003. Similarly should be done in other Tables and Figs. I would suggest presenting exact  P-values not * and indicate in the table /Fig legend the P-value under which results are considered significant.

Respond: We are very sorry for our poor statistic level. According to reviewer’s advice, we refer to previous study by Yu et al.[1]. All P values of the associations or interactions with FAs or minerals were corrected for multiple testing by false discovery rate method (FDR). PFDR(i) =p(i) * length(p) / rank(p). The tables and figures were revized. P and PFDR after FDR correction were presented in table 2, 3, 4 and 5. PFDR< 0.05 was noted in fiure 2 and 3. In addition, supplement 1 and 2 were added to present the P and PFDR in partial correlation analysis of mineral elements with erythrocyte FAs in children according to P50 stratification of whole blood Cu or Zn status.

It is also importnat to stress that all correlation coefficients are low or very low. This in combination with corrections for multiple comparisons which may indicate that some correlations are not significant can markedly change the conclusion of the paper.

Respond: We are extremely appreciative of reviewer’s this advice. Some unreliable results were corrected after corrections for multiple comparisons. In our previous results, significant inverse correlations of whole blood Ca with C22:6n-3 and total n-3PUFAs were observed by correlation analysis. Mg was negatively associated with C18:0, total SFAs, and C18:2n-6. there were low or very low correlation coefficients. However, no significant correlations were found after FDR correction. More reliable conclusions were presented in current study.

The authors concluded that their results provide new biomarkers. This is not possible to conclude based on the very week correlations and without corrections for multiple comparisons.

Respond: This improper expression had been removed according to reviewer’s advice.

Why the study is called cross-sectional?

Respond: We are very sorry for our inappropriate description. Cross-sectional study is to collect and describe the distribution of diseases or health status and related factors in a specific time point and a specific range of people, so as to provide clues for further research. This study was conducted in a single primary school in 2019 in Xiangtan, Hunan Province, located in South Central China. 435 children aged 4–7 years were recruited. Whole blood mineral elements were determined by atomic absorption spectrometry and erythrocyte FAs composition by gas chromatography-mass spectrometer. The relationships between erythrocyte fatty acids (FAs) levels with whole blood mineral elements concentrations were explored. Therefore the cross-sectional design was considered in this study.

Lane 261: Figure 3 should be Figure 4. Please correct.

Respond: Figure 3 had been corrected to Figure 4 in current manuscript.

  1. Yu, X.; Xia, L.; Zhang, S.; Zhou, G.; Li, Y.; Liu, H.; Hou, C.; Zhao, Q.; Dong, L.; Cui, Y., et al. Fluoride exposure and children's intelligence: Gene-environment interaction based on SNP-set, gene and pathway analysis, using a case-control design based on a cross-sectional study. Environ Int 2021, 155, 106681, doi:10.1016/j.envint.2021.106681.

 We tried our best to improve the manuscript and made some changes in the manuscript. We appreciate for Editors and Reviewers’ warm work earnestly, and hope that the correction will meet with approval. Once again, thank you very much for your comments and suggestions.

Yours

Sincerely

Huang Zhi
